# Anoctamins and Calcium Signalling: An Obstacle to EGFR Targeted Therapy in Glioblastoma?

**DOI:** 10.3390/cancers14235932

**Published:** 2022-11-30

**Authors:** Brittany Dewdney, Lauren Ursich, Emily V. Fletcher, Terrance G. Johns

**Affiliations:** 1Cancer Centre, Telethon Kids Institute, Perth, WA 6009, Australia; 2Centre for Child Health Research, University of Western Australia, Perth, WA 6009, Australia; 3School of Biomedical Sciences, University of Western Australia, Perth, WA 6009, Australia

**Keywords:** glioblastoma, high-grade glioma, EGFR, anoctamins, ion channels, calcium signalling

## Abstract

**Simple Summary:**

Glioblastoma is the most lethal form of brain cancer in adults. No new successful treatments have been developed in 30 years and survival rates have not improved, primarily because of a lack of effective drug treatments. Up to 60% of glioblastoma tumours have increased activity of a growth factor called epidermal growth factor receptor, which drives tumour growth. However, targeted therapies against the epidermal growth factor receptor have failed in clinical trials. A key reason for this is cell plasticity, a trait of brain cells that allow them to change their function in response to their environment. Tumour cells use plasticity to evade anti-cancer drugs. A group of genes called anoctamins may be involved in promoting tumour cell plasticity, which are believed to regulate cancer cell behaviour. This review summarises how anoctamins may regulate growth factor signalling and discusses a novel theory on how anoctamins may contribute to treatment resistance in glioblastoma.

**Abstract:**

Glioblastoma is the most common form of high-grade glioma in adults and has a poor survival rate with very limited treatment options. There have been no significant advancements in glioblastoma treatment in over 30 years. Epidermal growth factor receptor is upregulated in most glioblastoma tumours and, therefore, has been a drug target in recent targeted therapy clinical trials. However, while many inhibitors and antibodies for epidermal growth factor receptor have demonstrated promising anti-tumour effects in preclinical models, they have failed to improve outcomes for glioblastoma patients in clinical trials. This is likely due to the highly plastic nature of glioblastoma tumours, which results in therapeutic resistance. Ion channels are instrumental in the development of many cancers and may regulate cellular plasticity in glioblastoma. This review will explore the potential involvement of a class of calcium-activated chloride channels called anoctamins in brain cancer. We will also discuss the integrated role of calcium channels and anoctamins in regulating calcium-mediated signalling pathways, such as epidermal growth factor signalling, to promote brain cancer cell growth and migration.

## 1. Introduction

Gliomas are the most common primary malignant brain tumour, with the highest incidence rates in countries with higher socioeconomic status, including Australia. Several studies have reported Australia having the highest age-adjusted glioma incidence rates [1,2]. Glioblastoma multiforme (GBM), a lethal, grade IV astrocytic glioma, is the most common form of malignant glioma in adults [3,4,5]. In Australia, the incidence of GBM has increased, reaching an average of 3.4 cases per 100,000 person years [6]. GBM prognosis is one of the worst among all human cancers. The median survival time of GBM patients is usually less than two years. With treatment, the 2-year survival rate ranges from 20–35%, and the 5-year survival rate is approximately 10%, depending on patient age and therapies used [7,8,9,10,11]. The standard first-line treatment for newly diagnosed GBM patients is surgical resection followed by radiation therapy plus concurrent and adjuvant temozolomide (TMZ) treatment [11,12,13]. Although post-surgical administration of TMZ is important for increasing the overall survival of newly diagnosed GBM patients [14,15], the use of alternating electric fields to disrupt tumour cell mitosis as an adjuvant therapy in combination with TMZ may further improve survival [16,17,18]. Even still, tumour reoccurrence often occurs within a year due to the highly invasive nature of GBM and the infiltrating tumour cells within the surrounding tissue. There is no standard approach for second-line treatment in recurrent GBM, and the treatment course is determined on a case-by-case basis. Treatment options include reirradiation, systemic therapies, and/or reoperation, in which the extent of resection at reoperation becomes a significant prognostic factor in patient survival [19]. However, even with optimal treatment practices, GBM remains one of the deadliest diagnoses in modern-day oncology.

Most GBM tumours are isocitrate dehydrogenase (IDH) wild-type, which is often used to distinguish them from grade II and grade III oligodendrogliomas and other astrocytomas with mutated IDH. The presence of wild-type IDH alongside other specific genetic alterations, such as chromosomal aberrations (gain of 7p and loss of 10q), telomerase reverse transcriptase mutation, and epidermal growth factor receptor (EGFR) amplification/mutation, are most commonly present in grade IV GBM [20]. EGFR is amplified in up to 60% of GBM cases, with most of these tumours also harbouring EGFR deletion and point mutations [21,22]. EGFR status can not only be used diagnostically, but also as a potential therapeutic target in GBM. Despite the extensive research that has been conducted on the use of small molecule inhibitors for EGFR and relevant downstream signalling pathways, little evidence supports improved GBM patient outcomes, compared with standard TMZ treatment [20,23,24]. Therefore, there is an unmet clinical need for improved therapeutics, particularly in patient-specific treatment, to provide the best possible patient outcomes and overall survival. 

Active research aims to understand the mechanisms behind resistance to EGFR-targeted inhibition by investigating the dynamic interrelationship between EGFR signalling and other tumorigenic pathways. Although still in the early stages of investigation, one novel avenue is the role of ion channels in tumour cell signalling. This review will summarise the role of EGFR in cancer cell biology and the current challenges with EGFR-targeted therapy. Furthermore, this review will explore the role of calcium-mediated ion channels in tumorigenesis and discuss whether the link between EGFR and ion channel signalling could provide an opportunity to develop new targeted therapies for GBM patients.

## 2. EGFR Signalling and Variant Expression in GBM

EGFR (ErbB-1/HER-1), along with ErbB-2 (HER-2), ErbB-3 (HER-3), and ErbB-4 (HER4), belong to the family of ErbB receptor tyrosine kinases [25,26]. ErbB proteins contain a conserved intracellular domain, which includes the tyrosine kinase domain, a single pass hydrophobic transmembrane domain, and a less conserved extracellular domain responsible for ligand binding [27]. EGFR binds primarily to EGF, transforming growth factor α, and amphiregulin. Other ligands, such as betacellulin, heparin-binding growth factor, and epiregulin, may bind to both EGFR and ErbB-4 receptors [26]. Ligand binding stimulates receptor homo- and heterodimerisation. ErbB-2 is the preferred receptor for heterodimerisation because it primarily functions as a co-receptor and is incapable of ligand binding [27]. Dimerisation activates and phosphorylates the tyrosine kinase domains on the intracellular tail, providing a docking site for signalling proteins containing Src homology 2 domains or phosphotyrosine binding domains [25,26]. The main downstream signalling proteins include growth factor receptor-bound protein 2/son of sevenless (Grb2/Sos), phosphoinositide phospholipase C (PLC), phosphoinositide 3-kinase (PI3K), and janus kinase (JAK). The Grb2/Sos complex activates the Ras/Raf/MEK/ERK signalling cascade. PLC metabolises phosphoinositol complexes to form inositol 1,4,5-trisphosphate (IP_3_), which stimulates endoplasmic reticulum (ER) calcium store release and activates protein kinase C (PKC). PKC is well known to be involved in promoting GBM cell migration and invasion [28,29,30]. PI3K inhibits apoptosis and promotes cell survival by activating protein kinase B (Akt) signalling through phosphatidylinositol 4,5-bisphosphate/phosphatidylinositol-3,4,5-triphosphate (PIP2/PIP3) formation. JAK interacts with EGFR to phosphorylate and activate STAT transcription factor proteins. The roles of these pathways in regulating cancer cell proliferation, apoptosis, and cell stemness are well established and have been covered extensively in other reviews [31,32,33]. 

Up to 60% of GBM tumours have EGFR amplification and/or mutation [21,22,34]. A large number of the EGFR variants found in GBM are characterised by a deletion in the extracellular or intracellular domain [34]. The most common EGFR deletion is EGFRvIII, which lacks exons 2–7 in the extracellular domain and is found in as much as 66% of GBM tumours with amplified EGFR [22,35,36,37,38,39,40,41,42]. While EGFRvIII is incapable of ligand binding, both EGFRvIII and another variant, EGFRvII (exon 14–15 deletion), are constitutively active [43]. Other EGFR mutants have been identified, including deletions in the cytoplasmic tail (exon 25–27 deletion and exon 25–28 deletion), EGFR genomic rearrangements, and extracellular domain point mutations, although these variants are expressed at lower frequencies [21,35]. These variants provide a unique oncogenic advantage to tumour cell signalling and GBM survival [34]. However, EGFRvIII expression remains the most common and problematic EGFR mutation in GBM. 

Although EGFRvIII expression is specific to tumour cells and is not found in healthy tissue, its expression is heterogenous within the GBM tumour, and can be lost upon GBM recurrence [43,44,45,46]. This heterogeneity can be attributed to several factors. First, EGFRvIII-expressing cells drive the growth of EGFR amplified cells through secretion of cytokines and growth factors, thus there is not a strong selection for EGFRvIII cells in the whole tumour [44]. Second, development of the EGFRvIII mutation is a late event, so many other clones are already established before its appearance [44]. Finally, EGFR amplicons and the EGFRvIII variant may also be found on extrachromosomal double minute DNA fragments, which may be reversibly lost as a mechanism of therapeutic resistance [47,48]. Unequal segregation of extrachromosomal double minute DNA fragments containing EGFRvIII may further contribute to the heterogenous expression within the tumour. Collectively, these unique and dynamic expression patterns make EGFR-targeted therapy in GBM a challenging task. 

## 3. EGFR Inhibition in GBM–Clinical Trials and Limitations

### 3.1. EGFR Small Molecule Inhibitors

Several EGFR-targeting drugs, including small molecule inhibitors, antibodies, and vaccines, have been trialled in newly diagnosed and recurrent GBM patients, but with very limited success (Table 1) [31,49]. First-generation small molecule EGFR inhibitors (gefitinib, erlotinib, and lapatinib) block the ATP site of the tyrosine kinase domain and have reasonable tolerability in GBM patients, but do not increase overall survival [50,51,52,53,54,55,56,57,58,59,60,61,62,63]. Significant limitations of these first-generation inhibitors are their limited ability to maintain occupancy of the ATP-binding site on EGFR and their lower affinity for mutant forms such as EGFRvIII [64]. The second-generation inhibitors, afatinib and dacomitinib, were designed to irreversibly bind to the kinase domain of both EGFR and other ErbB family members but also failed to improve clinical outcomes as a single agent therapy in recurrent GBM patients [65,66,67]. A third-generation EGFR inhibitor, osimertinib, is the newest FDA-approved small molecule EGFR inhibitor and binds to the cysteine 757 residue in the ATP pocket of EGFR [68]. Promising preclinical studies show that osimertinib can efficiently cross the blood–brain barrier and inhibit both wildtype EGFR and EGFRvIII [69,70]. A case study of a woman with multifocal GBM demonstrated a complete response of one of the tumour sites to osimertinib treatment; however, the disease progressed at the other tumour site, likely due to the heterogeneous nature of the patient’s EGFR mutation status [71]. A phase II clinical trial of osimertinib is currently underway for patients with recurrent GBM (NCT03732352).

### 3.2. Anti-EGFR Antibodies and Antibody-Drug Conjugates

FDA-approved monoclonal anti-EGFR antibodies, including cetuximab, panitumumab, and nimotuzumab, bind to the extracellular L2 domain of the EGFR ligand binding site, thereby preventing ligand binding and receptor dimerisation. Although these antibodies have shown success in other cancer types, to date, they have shown little efficacy in glioma patients [83]. Cetuximab has failed to improve outcomes in EGFR-expressing GBM patients compared with non-expressing patients [72,73]. Furthermore, a preclinical study has shown that cetuximab inhibits phosphorylation of wild-type EGFR and causes internalisation of EGFRvIII but does not inhibit EGFRvIII activity. Instead, it causes increased EGFRvIII phosphorylation and downstream signalling, thus leading to drug resistance [84]. Similarly, clinical trials have shown that nimotuzumab treatment is well tolerated and may improve survival outcomes in newly diagnosed GBM patients, but these survival rates were not significantly higher than the control groups treated with standard radiochemotherapy alone [74,75,76]. Additionally, nimotuzumab has been trialled in paediatric high-grade glioma (HGG) as a tolerable and potential treatment option for newly diagnosed or relapsed/progressing HGG patients, but with no significant improvements in HGG patient survival [85,86]. A preclinical study demonstrated that of all the clinically approved anti-EGFR antibodies, panitumumab was the only one capable of neutralising both EGFR and EGFRvIII in vitro and in vivo [87].

Other antibody agents have been developed with increased specificity in targeting EGFR and EGFRvIII, the latter being nonresponsive to the antibodies described above due to the deleted L2 domain. Depatuxizumab mafodotin (Depatux-M) is an antibody-drug conjugate derived from the monoclonal antibody mAb 806, which selectively targets EGFR amplification and EGFRvIII specifically in GBM tissue [88]. The antibody is conjugated to monomethyl auristatin F, a cytotoxin that inhibits microtubule function and results in cell cycle arrest and apoptosis [77]. In a recent phase II clinical trial (Intellance2), Depatux-M and TMZ combination treatment significantly increased 2-year survival rates in a subset of GBM patients who relapsed 16 weeks after completion of TMZ treatment compared with those treated with Depatux-M alone or TMZ alone. However, the primary endpoint of improved overall survival for the entire patient cohort was not reached, despite a trend towards increased survival in the Depatux-M plus TMZ-treated patients [78]. Another single-arm multicentre observational study of recurrent GBM patients showed similar survival outcomes in patients treated with Depatux-M plus TMZ [79]. A phase II trial (Intellance1) is currently underway to determine the efficacy of Depatux-M in newly diagnosed GBM patients (NCT02573324).

### 3.3. Other EGFR-Targeted Agents

Rindopepimut is a vaccine consisting of an EGFRvIII peptide sequence designed to activate a specific immune response towards EGFRvIII-expressing tumour cells [89]. A phase II clinical trial showed promising results with loss of EGFRvIII expression in most newly diagnosed GBM patients treated with rindopepimut alongside standard radiochemotherapy treatment [80]. However, in the phase III clinical trial, there was no significant difference in overall survival of newly diagnosed GBM patients treated with the vaccine and TMZ compared with those treated with the vehicle control and TMZ [81]. 

Chimeric antigen receptor (CAR) T cells have also been investigated for their potential benefit in GBM, particularly in targeting EGFRvIII, which is known to be a tumour-specific epitope. Although preclinical studies demonstrated this was a promising approach [90,91], in a pilot study of 10 GBM patients treated with EGFRvIII-targeting CAR T cells, no objective responses were met [82]. A new clinical trial is currently underway combining CAR T cell therapy with mAb806 to target amplified EGFR and EGFRvIII with more tumour specificity. This trial is currently recruiting children and young adults with various non-central nervous system solid tumours to evaluate this therapy’s tumour-specific and non-specific off-target effects (NCT03618381). Once maximum tolerated doses and safety profiles are established, this therapy may move to future GBM clinical trials.

## 4. The Role of EGFR in Cancer Cell Growth, Invasion, and Survival in GBM

Despite these extensive attempts to improve survival outcomes for GBM patients with EGFR-expressing tumours, there is little clinical evidence supporting the benefit of EGFR-targeted therapies in GBM treatment. The heterogenous nature of GBM certainly contributes to the treatment difficulties [92]. Other genetic aberrations in genes such as platelet-derived growth factor receptor, cyclin dependent kinase 4, and neurofibromin are also involved in regulating progenitor and mesenchymal-like cell states within the heterogenous tumour [93]. In fact, GBM tumours can be derived from a heterogenous mix of these different cell states, demonstrating the highly plastic nature of GBM cancer cells [93]. More specifically, high EGFR expression is related to maintaining a highly proliferative, astrocytic-like state in GBM tumour cells [93]. In this regard, one of the main reasons for therapeutic resistance in GBM is due to the proliferative and highly invasive nature of GBM tumours, which can be attributed to cellular plasticity and the dynamic changes in expression and signalling of EGFR. 

### EGFR and Cell Migration and Invasion: Extracellular Matrix Remodelling and Intracellular Signalling

The earliest steps in growth factor-mediated cell migration involve the co-localisation of integrins with EGFR at focal adhesions upon growth factor stimulation [94]. Integrins are membrane bound adhesion molecules that bind to the extracellular matrix (ECM) and are required for cell migration. Co-localisation of integrins and EGFR at the focal adhesion site is followed by recruitment of the focal adhesion kinase (FAK)-Src complex to facilitate downstream signalling for cell migration, especially by activating the Rho GTPase family, which regulates actin for cytoskeleton remodelling required for cell migration, adhesion, and polarity [95,96]. Of course, EGFRvIII may also act as a scaffold to activate FAK to promote glioma cell migration [97,98,99]. Additionally, the signalling cascades associated with FAK-mediated cell migration and EGFR cell signalling significantly overlap, as FAK may also activate the Ras/ERK and PI3K/Akt cascades [94,100,101], demonstrating the complexity of EGFR signalling in cancer cells depending on the context of activation and localisation on the cell itself. 

Cell migration is required for GBM cells to invade through the ECM and infiltrate through the surrounding tissue. It has been well characterised that EGFR and FAK activation promote cell invasion through increased expression of matrix metalloproteinases (MMP), particularly MMP-2 (gelatinase-A) and MMP-9 (gelatinase-B) [99,102,103,104], as well as other proteases [105], which are proteolytic enzymes that degrade proteins within the ECM to facilitate tumour cell migration and invasion. GBM tumours are frequently found to have upregulation of MMP-2 and MMP-9 [106,107,108,109]. Indeed, active MMP-9 expression can occur in up to 73% of EGFR amplified tumours and is strongly correlated with EGFRvIII expression, with MMP-9 expression occurring in up to 83% of EGFRvIII-expressing tumours [110]. The precise signalling pathways involved in EGFR/FAK-mediated cell invasion are unclear, as it appears that all the main downstream effectors, including ERK [111,112,113], Akt [104,114], and STAT [99,103,115], may play a role in activating MMP expression. A recent study demonstrated that EGF-mediated activation of MMP-9 occurred through increased PI3K/Akt, ERK1/2, and STAT3/STAT5 signalling in glioma cells, leading to NF-κB localisation to the promoter of the *MMP-9* gene and promoting GBM migration and invasion in vitro [116]. Furthermore, EGFRvIII has been shown to be critical for activation of FAK through the JAK2/STAT3 axis required for MMP-2/9 expression and GBM cell invasion [97,99]. 

However, the signalling cascades associated with EGFR activation and tumour cell growth and invasion are complex and have yet to be fully elucidated. For example, there are apparent differences in the downstream signalling pathways of EGFR and EGFRvIII in GBM cells. Lorimer and Lavictoire elegantly demonstrated that EGFRvIII promoted constitutive phosphorylation of ERK and Akt, and that this activity was dependent on P13K kinase activity [117]. Furthermore, inhibition of PLC also blocked ERK phosphorylation by EGFRvIII. In contrast, EGF-mediated activation of wild-type EGFR resulted in phosphorylation of ERK and Akt even in the presence of P13K inhibitors and PLC inhibitors, suggesting that wild-type EGFR utilises different kinase signalling cascades to EGFRvIII [117]. Notably, EGFRvIII-expressing glioma cells may promote increased expression of PKC [28], the downstream effector enzyme of PLC, which is known to be a modulator of the ERK signalling cascade and is involved in glioma cell migration [118,119]. This is also supported by proteomics data which has shown that EGFRvIII-expressing glioma cells secrete greater numbers and levels of invasion-promoting proteins compared to wild-type EGFR glioma cells, such as increased expression of MMP-2 and the protease cathepsin B [120], both of which are responsible for ECM remodelling and cell invasion. It is clear that GBM cell proliferation, migration, and invasion can be achieved through multiple signalling cascades, activated by kinase activity of wild-type EGFR, EGFRvIII, or both, highlighting the multifarious nature of EGFR signalling in GBM, which makes targeting EGFR as an anti-tumour strategy challenging for translation into clinical benefit. 

To further complicate the dynamics of EGFR in cancer cell biology, EGFR has a variety of kinase-independent (KID) pro-survival roles in cancer cells [121]. The ability of EGFR to stabilise the sodium/glucose cotransporter 1 (SGLT1) to maintain glucose levels and prevent autophagy death in a KID manner, was the first detailed study describing this phenomenon [122]. Similar KID interactions with fatty acid synthase (FASN) and PUMA all promote pro-survival in cancer cells [121]. KID-EGFR activity may also result from dimerisation with other ErbB family members or through the cross-talk of other kinases, such as the mammalian target of rapamycin (mTORC2) [121,123]. The KID activity of EGFR is another possible mechanism by which EGFR inhibitors are less efficacious than expected in a range of tumours. 

One aspect of tumour biology that may also drive therapeutic resistance in GBM is cellular plasticity [92]. Maintaining ionic currents is essential for cellular signalling in normal astrocyte function and EGFR-amplified tumours. This mechanism could regulate cancer cell plasticity. Thus, EGFR+ astrocyte-like GBM is likely to have dysregulated ion channel expression that mirrors the highly proliferative state of the astrocyte-like cells. Ion channels regulate cell proliferation, cell migration, and tumour invasion by controlling the movement of important ions, such as calcium, potassium, and chloride, that act as signals for cellular pathways and cell volume regulation [124,125]. Calcium waves, known to modulate astrocyte processes and contribute to neuronal plasticity [126], may be especially important in astrocytic-like GBM. Calcium waves, along with EGFR signalling, which is also dependent on calcium signalling, may support tumour cell proliferation, migration, and invasion. However, the physiological mechanisms of how ion channels contribute to GBM progression remains a relatively unexplored research avenue. The remainder of this review will focus on a few classes of ion channels involved with calcium signalling, including a family of calcium-activated chloride channels (CaCC) called anoctamins (ANOs), which have been implicated in cancer pathogenesis in recent years. 

## 5. ANOs in Cancer

The ANO family contains 10 protein members, ANO1–ANO10 (TMEM16A–TMEM16K) [127,128]. ANO1 and ANO2 are well studied and have been shown to have calcium-dependent chloride ion channel activity [129,130,131,132]. However, other members of the ANO family are less well characterised and may have other functions. The activity and selectivity of ANOs largely depend on intracellular Ca^2+^ concentrations and on the proliferative state of the cell. For example, proliferating cells may have increased expression of ANO1, ANO6, and ANO10 [127]. Several recent studies have shown that ANO1 (TMEM16A) is overexpressed in many cancer types such as breast cancer, head and neck squamous cell carcinoma (HNSCC), prostate cancer, colorectal cancer, and glioma (Table 2) [133]. 

### 5.1. ANOs and Cell Migration: Regulators of Cell Volume

ANOs may regulate cancer cell volume changes by increasing chloride efflux and causing cell shrinkage during cancer cell migration [150,151]. Increasing evidence demonstrates the pivotal role of ion channels, particularly Cl^−^ channels, in the invasion and migration of glioma cells [152]. Cells must adapt their shape and volume when migrating through narrow pores. This is achieved through net ion uptake and efflux and local regulated volume decrease (RVD) [153,154]. Several ANO paralogues are activated during hypoosmotic cell swelling, stimulating Cl^−^ conductance and RVD [151,155]. The membranous ion channels and proteins involved in osmotic water flux often display altered activity or expression in metastatic cancer cells [156]. For example, ANO1 expression is required to initiate swelling-activated whole-cell currents in colonic epithelium and salivary acinar cells in vivo [151]. Furthermore, ANO1-generated Cl^−^ efflux accompanies potassium (K^+^) efflux and collectively drives the osmotically-determined H_2_O efflux necessary for RVD [157]. Additionally, knockdown of endogenous ANO6, ANO8, and ANO10 in vitro also reduces volume-regulated chloride currents, demonstrating the potential involvement of other ANOs in cell volume regulation [151].

### 5.2. ANO1 in Glioma

ANO1 is expressed in glioma cancer cell lines and patient glioma tissues, and expression correlates with a high pathological grade [133,137,142,158,159]. Furthermore, several studies have shown that ANO1 knockdown reduces cell proliferation, migration, and invasion of glioma cells in vitro, suggesting a critical role in brain cancer cell function [137,142,159]. Although the precise mechanisms of ANO1 in glioma progression remain unclear, it was first suggested that ANO1 expression is associated with activation of the nuclear factor-κB (NF-κB) signalling pathway [137]. More recently it has been shown that ANO1 directly interacts with and stabilises EGFRvIII in glioma cells [142]. Importantly, the oncogenic function and constitutive activity of EGFRvIII is largely dependent on its stability within the plasma membrane, as determined by its ability to continuously trans-phosphorylate itself and resist internalization and degradation [160,161,162]. Furthermore, inhibition of ANO1, through knockdown or drug inhibition, reduces the expression of EGFRvIII, NOTCH1, and other cancer stem cell markers [142]. ANO1 has also been shown to mediate EGFR signalling in HNSCC and breast cancer cells [136,139]. This evidence suggests that ANO1 expression may play a significant role in promoting the activation of various cell surface receptors on cancer cells that are required for intracellular signalling and cell growth. However, it has yet to be determined whether ANO1 also binds with NOTCH receptors in a similar manner to EGFR in glioma.

### 5.3. Other ANOs in Cancer

ANO6 is believed to act both as a lipid scramblase and a nonselective ion channel [128,163]. ANO6 is involved in regulating cell viability and apoptosis via calcium-mediated lipid scrambling and externalisation of phosphatidylserine [163,164,165,166]. ANO6 knockdown impairs cell migration in Ehrlich-Lettre ascites cells along with ANO1 [167]. ANO6 also supports calcium-mediated RVD [165], which plays a coordinated role in cell migration and proliferation [168,169]. A recent study showed that ANO6 is increased in human glioma tissue and that high ANO6 expression is associated with worsened survival (Table 2) [146]. Furthermore, ANO6 knockdown reduces glioma cell viability, proliferation, and invasion by inhibiting ERK signalling [146]. However, it remains unclear whether ANO6 regulates glioma cell progression via ion current activity, scramblase activity, or a combination of these effects in synergy with other ANO proteins such as ANO1. 

ANO9 has been studied for its role in pancreatic cancer where, like ANO1, it binds to and activates EGFR and downstream ERK signalling (Table 2) [148]. EGFR forms complexes with both ANO1 and ANO9, but EGFR preferentially binds to ANO9 in much higher quantities than ANO1 [148]. Consequently, ANO9 promotes pancreatic cancer cell proliferation in vitro and in vivo [148]. Furthermore, ANO9 knockdown sensitises pancreatic cancer cells to EGFR-targeted treatment with erlotinib [148]. Similarly, ANO5 has also been studied for its role in proliferation and migration in pancreatic cancer [143], thyroid cancer [144], and osteosarcoma [145] models in vitro, and ANO7 has been associated with poor prognosis in prostate cancer [147] (Table 2). This suggests that other ANOs besides ANO1 may play similar roles in cancer cell progression. However, ANO pathways in brain cancer are still largely unexplored.

## 6. ANO-Targeted Therapy in Cancer

ANO1 has become a therapeutic target of interest, and several small molecule inhibitors have been shown to reduce ANO1 activity (Table 3). For example, CaCC_inh_-A01 is a non-selective CaCC inhibitor that reduces ANO1 current activity, causes ANO1 protein degradation, and inhibits cell proliferation and invasion [142]. CaCC_inh_-A01 also inhibits the growth of HNSCC cells by reducing EGFR activity and sensitises the cancer cells to EGFR-targeted therapy [139,170]. Similarly, another CaCC inhibitor, diethylstilbestrol (DES), also inhibits ANO1 and ANO2, reduces EGFR activation, and decreases non-small cell lung cancer (NSCLC) cell migration [171]. However, considering that CaCC_inh_-A01 and DES are non-selective CaCC inhibitors, it cannot be concluded that the reduced EGFR signalling and anti-cancer effects are due to reduced ANO1 current activity. It has yet to be explored whether CaCC inhibitors alongside EGFR-targeted therapy would be a promising combination therapy in glioma. 

More recent studies have assessed the effects of more specific ANO1 inhibitors. Seo and colleagues identified ‘3n’, a compound derived from a 2,2-dimethyl-2G-chromene motif, that had increased selectivity for inhibiting ANO1, with only weak inhibition of ANO2 [173]. Furthermore, this study showed that 3n induced apoptotic cell death in prostate cancer cells in vitro [173]. Another study identified two compounds derived from 2-aminothiophene-3-carboxamide that inhibit ANO1 current activity to a similar extent to CaCC_inh_-A01 but without suppressing the activity of other chloride channels [174]. These two compounds, (6-(tert-Butyl)-2-pivalamido-4,5,6,7-tetrahydrobenzo[b]thiophene-3-carboxamide and 2-(3-(4-Chlorobenzoyl)thioureido)-5,6-dihydro-4H-cyclopenta[b]thiophene-3-carboxamide), proved to be effective anti-glioma agents and had more potent effects on reducing glioma cell proliferation, migration, and invasion than CaCC_inh_-A01 [174]. In addition, when the former compound was combined with TMZ, glioma cell proliferation was significantly reduced compared with either treatment alone, demonstrating the potential for ANO inhibition as a novel avenue for targeted therapy in GBM [174].

Several other generic CaCC inhibitors have been identified, such as cepharanthine, niclosamide, tannic acid, niflumic acid (NFA), and 5-nitro-2-(3 phenylpropylalanine) benzoate (NPPB) (Table 3). The ability of these inhibitors to block chloride ion currents has been explored [190,191,192]. However, these inhibitors often lack specificity. NFA and NPPB are more traditional CaCC inhibitors that may inhibit a range of chloride channels, including ANO1, ANO2, and ANO6, as well as other ion channels, including cation channels [193,194,195,196,197,198]. Both have anti-proliferative activities in cancer models in vitro [176,177,178], although this cannot be attributed to ANO inhibition alone. Cepharanthine is more selective for ANO1, but also inhibits ANO2 channel activity and weakly inhibits ANO6 activity [175]. Cepharanthine reduces lung adenocarcinoma growth in vitro and in vivo [175]. Tannic acid is slightly more specific for ANO1/ANO2 and is a strong inhibitor of both ANO1 and ANO2 currents [190], but may only indirectly reduce ANO6 currents without any effect on scramblase activity [195,199]. Nonetheless, gallotannin induced apoptosis and cell cycle arrest in preclinical studies of colon, breast, prostate, and liver cancer cells [184,185,186,187,188,189]. More recently, enriched gallotannin, extracted directly from the gall of the *Quercus infectoria* plant, showed comparable cytotoxic effects on GBM cell viability to traditional TMZ, but with more potent antioxidant properties [183]. Furthermore, commercially available synthetic gallotannin was significantly less effective at reducing GBM cell viability, suggesting that the naturally bioactive compound derived directly from the plant itself has superior cytotoxic effects on cancer growth [183]. This phenomenon remains to be explored in other aspects of GBM pathogenesis, such as migration and invasion, cell signalling, and tumour growth in in vivo preclinical models. There are no currently registered clinical trials investigating gallotannin as a treatment for any human cancers. Furthermore, no studies have explored whether the antineoplastic effects of gallotannin in cancer cells are related to ANO channel function. Interestingly, one study has shown that gallic acid, the polyphenol monomer that is found in gallotannin, may inhibit EGFR signalling and MMP-9 activation in breast cancer cells [200]. Thus, gallotannin may also have an inhibitory effect on the signalling pathways associated with cancer cell migration and invasion, although this remains to be fully elucidated.

Niclosamide, an oral FDA-approved drug that has been used to treat parasitic infections for many decades, is a potent ANO1 inhibitor, and also inhibits ANO6 currents [192,201,202,203]. More recently, this compound has been repurposed for other medical conditions, such as cancer, as it inhibits multiple tumour cell signalling pathways in a wide range of human cancer cells, including GBM cells [180,182,204]. Niclosamide has been evaluated in a phase I clinical trial for its safety in combination with traditional enzalutamide therapy in prostate cancer patients (NCT02532114). However, as determined by preclinical studies, the maximum tolerated dose did not result in plasma concentrations reaching the optimal effective concentration for drug efficacy, suggesting that niclosamide has poor oral bioavailability [205]. Increasing niclosamide dose resulted in significant toxicities, and this combination treatment was not investigated further as a prostate cancer treatment [205]. However, a more recent phase 1b clinical trial showed that a newly formulated niclosamide compound has increased bioavailability in prostate cancer patients [206]. The encapsulated niclosamide (PDMX1001) showed no dose-limiting toxicities and patients were able to reach the therapeutic plasma concentration with tolerable adverse events as part of a drug combination cocktail with abiraterone acetate and prednisone [206]. This therapeutic combination is now being investigated in a phase II clinical trial (NCT02807805). PDMX1001 is also being trialled with the traditional enzalutamide treatment (NCT03123978) to determine whether the change in formulation or drug combination improved the bioavailability and toxicity profiles of niclosamide. Niclosamide has also briefly been tested in several patients with metastatic colorectal cancer in the NIKOLO trial. Preliminary results showed that high plasma levels of niclosamide was achievable with no dose-limiting toxicities in patients [207]. The NIKOLO trial has moved into a phase II clinical assessment for metastatic colorectal cancer (NCT02519582); however, the status of this study remains unknown. Despite the promising preclinical evidence suggesting that niclosamide has antineoplastic activity in cancer cells, it has yet to be explored further in other clinical settings.

Preclinical studies have shown that GBM cells exposed to niclosamide simultaneously downregulate multiple cancer cell signalling pathways, including NOTCH, mTOR, MAPK/ERK, and Akt-dependent signalling, leading to reduced cancer cell proliferation, viability, and migration [180,182]. Interestingly, niclosamide inhibited NF-κB signalling only in primary GBM cell lines that had a heterozygous deletion of the NFKBIA locus (NFKBIA^+/−^) [180]. NFKBIA deletion is often associated with EGFR amplification in GBM patients and is associated with poor survival outcomes [208]. These results suggest that niclosamide may be an effective therapeutic compound to use in synergy with EGFR-targeted therapy for GBM patients with EGFR amplification and NFKBIA deletion. Similarly, niclosamide treatment combined with TMZ reduced GBM tumoursphere viability, stemness, and invasion [181]. However, like gallotannin, no studies have investigated whether the effect of niclosamide on blocking oncogenic signalling pathways is related to inhibited CaCC ion channel activity or another signalling mechanism. This intriguing area of research warrants further investigation to determine how ion channel function may relate to cell signalling pathways in cancer development and progression.

## 7. Calcium Channels and Their Relevance to ANO and EGFR-Related Signalling

The evidence linking ANO function to EGFR signalling in GBM is still in its infancy. The underlying mechanisms by which these two proteins act in synergy to maintain cellular plasticity in high grade gliomas has yet to be fully elucidated. What is known, however, is that intracellular calcium levels are critical, not only to normal cellular function, but also as significant secondary messengers for various oncogenic pathways. Calcium concentration largely influences the balance between a proliferative versus apoptotic state. Published data also suggests that EGFR-mediated calcium signalling is involved in drug resistance in cancer. In NSCLC cells, EGF signalling caused Ca^2+^ oscillations in non-resistant cells, whereas drug-resistant cells had lower ER calcium stores and increased extracellular Ca^2+^ influx [209]. Furthermore, restricting extracellular Ca^2+^ can induce drug sensitivity in NSCLC cells [209]. Considering what is known about increased EGFR and ANO expression in cancer, particularly in gliomas, further investigation into how calcium signalling may contribute to drug resistance and cellular plasticity in brain tumour progression is warranted. It would not be surprising if ion channels, such as ANOs, which are regulated by calcium, also contribute to drug resistance and cancer cell proliferation. Another more notable factor that should be considered is the underlying roles of other intra-membrane proteins that regulate flow of free Ca^2+^ between organelles and the cytoplasm, such as inositol 1,4,5-triphosphate receptor (IP_3_R) and transient receptor potential (TRP) channels. IP_3_R regulates the release of ER calcium stores, while TRP channels are primarily involved in Ca^2+^ influx through the plasma membrane [210]. Therefore, these channels may directly contribute to the activation of ANOs.

### 7.1. IP_3_R

IP_3_R is involved in the release of ER calcium stores into the cytosol and may be activated by a variety of stimuli; however, the most effective activation results from the presence of Ca^2+^ and IP_3_. This dual messenger system is particularly significant in neuronal cells, where even small increases in IP_3_ can increase the Ca^2+^ sensitivity of IP_3_R and may contribute to localised waves of Ca^2+^ signalling and cellular plasticity [211]. Furthermore, IP_3_R is not distributed evenly along the ER and may be found in clusters within the ER near the plasma membrane, generating “Ca^2+^ signalling microdomains” within the cell [212,213]. IP_3_R may also bind to plasma membrane proteins to tether the ER to the plasma membrane for localised calcium signalling.

ANOs are considered to act as membrane-bound ion channels activated by localised calcium signals. In nociceptive dorsal root ganglia neurons, ANO1 localises within lipid rafts in the plasma membrane and binds to IP_3_R to tether the plasma membrane to the ER. This creates an ANO1-specific microdomain and allows ANO1 activation via localised release of ER calcium [214]. Lipid rafts often provide signalling platforms for growth factor receptors, such as EGFR, in tumour development [215]. This is particularly relevant for EGFR signalling, which produces IP_3_ signals for ER calcium release and regulation of cancer cell growth. Further studies in other cell types have verified that interaction with IP_3_R activates ANO1 through IP_3_R-mediated calcium release [216,217].

### 7.2. TRP Channels

TRP channels are a superfamily of 30 cationic cell membrane channels that are permeable to Ca^2+^ and directly activate intracellular Ca^2+^ signalling [210]. TRPC (TRP canonical) and TRPV (TRP vanilloid) are the two TRP channel subfamilies most commonly implicated in GBM [218]. A more comprehensive overview of TRP channels in brain cancer has been recently published; thus, this review will not cover this topic in great detail [218]. However, it is noteworthy that some of the TRP channels that have been investigated in GBM may play other unknown roles in EGFR/ANO signalling. For example, TRPV1 has been extensively studied for its role in regulating the balance between proliferation and apoptosis in various cancer cell lines; however, the evidence for it being pro-apoptotic or pro-proliferative is somewhat inconsistent [219]. Several studies have shown that TRPV1 binds and interacts with EGFR but it is unclear whether this interaction causes EGFR degradation and promotes apoptosis [220,221] or results in EGFR transactivation and promotes cell proliferation [222]. The role of TRPV1 in inducing apoptosis versus proliferation is likely dependent on the presence or absence of active EGFR and may also largely depend on the cell type. Most published data suggests that TRPV1 suppresses EGFR signalling in cancer cells and, therefore, has anti-tumorigenic effects [221,223,224]. In GBM specifically, TRPV1 loss correlates with higher tumour grade and shorter survival in patients [225,226]. Furthermore, TRPV1 upregulation and activation has a protective effect in glioma cell lines and induces apoptosis through increased intracellular calcium signalling [225,227,228]. Interestingly, TRPV1 is a known binding partner of ANO1 in the context of pain-enhancement mechanisms in neurons [229,230]. Whether TPV1/ANO1 interactions exist in a functional manner in GBM has yet to be determined.

In contrast, TRPV4 and TRPC1 are other TRP channels that have consistently been shown to play a pro-tumorigenic role in brain cancer. TRPV4 is upregulated in glioma, positively correlates with tumour grade in glioma patients, and is associated with a worse prognosis [231,232,233]. TRPV4 is another potential binding partner for EGFR [210,234,235], ANO1 [236,237], and IP3R [237], although this has yet to be demonstrated in brain cancer cells. Nonetheless, it has been shown that TRPV4 promotes glioma cell migration and invasion through Akt signalling and regulating the formation of cellular protrusions [231,233]. It could be postulated that while TRPV4 is important for the cytoskeletal changes associated with cell migration, an interaction with ANO1 might be critical in regulating cell volume changes to support glioma cell invasion. Furthermore, cancer cells with high TRPV4 expression are more sensitive to EGFR-targeted therapy, further suggesting that EGFR and TRPV4 expression may be functionally linked [238]. Similar to TRPV4, TRPC1 is also increased in GBM patients [239] and is involved in glioma cell proliferation [240,241] and migration via regulating chemotaxis at the leading edge of migratory glioma cells [242,243]. TRPC1 activation in response to EGF-induced chemotaxis regulates Cl^−^ channel activity in glioma cells in a calcium-dependent manner [243]. Although the effects of TRPC1 on ANO channels have not been investigated, it is possible that TRPC1 may also regulate or interact with calcium-activated chloride channels, such as ANOs. TRPC1 has also been shown to activate EGFR in NSCLC cells [244] but no studies have investigated its interactions with EGFR in GBM.

TRP channels have strong implications in cancer cell migration and invasion, particularly TRPM7, which is a member of the melastatin-related TRP family, and it has been found to be upregulated in GBM [239,245]. TRPM7 is one of the most highly characterised TRP channels, and it is known to have mechanosensory signalling abilities and kinase activity that regulates cellular contractility and motility [246]. Activation of TRPM7 and Ca^2+^ influx in GBM cells was associated with increased cell migration and invasion via activation of Akt [245], ERK1/2 [245,247] and JAK2/STAT3 [248] and increased MMP-2 expression [245,247]. It has been speculated that TRPM7 may phosphorylate and activate PLC to facilitate this downstream signalling cascade [245,249]. This observation raises the question if calcium-mediated PLC phosphorylation and activation can sustain intracellular signalling in an EGFR-independent manner. One study has identified that EGF increased expression and phosphorylation of TRPM7 in vascular smooth muscle cells, and that TRPM7 directly interacted with EGFR in a Src-dependent manner to promote cell migration [250]. There has yet to be any studies investigating the direct binding of TRPM7 and EGFR in cancer cells, nor is there any evidence of TRPM7 interacting with ANOs. Nonetheless, this highlights the significant knowledge gaps in our understanding of the cross talk between EGFR and ion channels for continued tumour progression.

### 7.3. Lipid Rafts Containing EGFR, ANOs, and Other Calcium Channels May Be Key to Cell Invasion, Cellular Plasticity, and Drug Resistance in GBM

Lipid rafts are localised regions within the plasma membrane of a cell that are enriched in cholesterol and sphingolipids and are often hyperactivated in cancer. Lipid rafts harbour various signalling proteins and ion channels known to be involved in proliferation and migration in cancer cells [251]. Several studies have demonstrated that EGFR is often localised within lipid rafts in cancer, including in GBM [215,252,253,254]. In fact, the cholesterol levels in the lipid raft are important for regulating EGFR activity [215,255,256] and cancer cells are more sensitive to EGFR-targeted therapy upon cholesterol depletion from lipid rafts due to increased phosphorylation levels [252,253]. Caveolae are lipid raft regions of the plasma membrane that, upon caveolin-1 integration, invaginate into flask-shaped membrane organelles. Similar to lipid rafts, caveolae may also serve as signalling platforms, particularly for EGFR [257], and they are also involved in activating clathrin-independent endocytosis upon phosphorylation of caveolin-1 [123,257]. There is controversial evidence on whether or not EGFR is concentrated within caveolae [254,256,258,259,260]. However, it is proposed that caveolin-1 binds to EGFR in caveolae after EGF stimulation, rendering it inactive, followed by internalisation of EGFR via endocytosis [254,258,261]. Notably, it has been shown that EGF-stimulated phosphorylation and activation of wild-type EGFR led to binding of caveolin-1 and dissociation from caveolae in GBM cells [254]. Furthermore, wild-type EGFR and caveolin-1 colocalised within lipid rafts, but EGFRvIII did not [254]. As EGFRvIII is constitutively phosphorylated, this observation is in parallel with other work that has demonstrated that depletion of lipids from lipid rafts increases EGFR phosphorylation [252,254,255,262]. Considering that EGFRvIII is constitutively phosphorylated in a ligand-independent manner, its localisation may not be specific to lipid rafts; rather, its stabilisation within the membrane [160], perhaps by ANO1 [142], is more significant to retain its oncogenic function. Interestingly, when GBM cells were treated with an EGFR inhibitor, caveolin-1 expression increased in both the wild-type EGFR and EGFRvIII-expressing cells. Thus, caveolin-1 interaction may be one mechanism in which EGFR-targeted therapies inhibit kinase activity, however, GBM cell survival is likely to continue due to stabilisation of EGFR expression and the presence of a now inactive, KID-EGFR. This is supported by previous work which demonstrated that EGFR inhibitors do not cause EGFR degradation [87].

Similarly, ANOs and TRP channels are also found in lipid rafts, often linked with receptor proteins such as EGFR as part of ion channel complexes in cancer cells, and function as a means of localising intracellular calcium signals [214,215,263,264]. Calcium signalling is imperative to tumour cell migration and invasion, where transient Ca^2+^ oscillations, waves, or flickers activate Ca^2+^-dependent effectors responsible for focal adhesion assembly and disassembly in the leading edge and retracting edge of the cell, respectively [265,266]. Migrating cancer cells exhibit this rear-to-front calcium gradient to support this phenomenon known as focal adhesion turnover [267]. Of course, tethering of the plasma membrane to the ER through ANO/IP_3_R interactions would also contribute to localised calcium signalling at the lipid raft junction [214,268]. Linking these concepts and evidence together, it is apparent that ANOs, particularly ANO1, may bridge the link between EGFR signalling and calcium signalling to regulate cancer cell proliferation, migration, and invasion (Figure 1). As observed with the SGLT1 transporter, we propose that EGFR may interact with ANOs, and other calcium channels, in both a kinase-dependent and KID manner to support their dynamic functions in regulating cell migration and invasion. Such a function would be impervious to the activity of EGFR inhibitors, thus, contributing to resistance to EGFR targeting drugs. In this speculative theory, the EGFR-ANO-TRP complex localised within lipid rafts may serve to stabilise wild-type EGFR or EGFRvIII expression and prevent internalisation. Such an interaction would allow sustained phosphorylation-dependent signalling through EGFR/EGFRvIII to enhance activation of cell migration and invasion pathways supported by localised calcium signalling and VRD through ANO-mediated Cl^−^ currents required for cell migration. Alternatively, these complexes may also serve to stabilise KID-EGFR expression within the membrane as a pro-survival mechanism. Remodelling of calcium signalling by altered expression of these key ion channels is likely to be the driving force behind cell migration and invasion in GBM and it would play a large contributing factor to drug resistance [246,269], particularly in EGFR-targeted therapies, as we have described that these calcium-dependent ion channels and regulators may themselves activate the signalling cascades associated with RTK activity. Although the papers discussed thus far have identified individual binding interactions between ANO1, EGFR, IP_3_R, and TRP channels, it is likely that multiple combinations of these interactions exist within several lipid raft domains along the plasma membrane of glioma cells, especially in the leading edge of migratory cells or during chemotaxis [233,243]. The dynamic nature of these interactions complicates our understanding of how ion channels contribute to tumour progression in brain cancer.

## 8. Future Directions and Conclusions

In recent years, it has become evident that drug resistance, particularly in the case of EGFR-targeted therapy, may be caused by dysregulated ion channel activity and intracellular calcium signalling. Indeed, EGFR-targeted therapy can rewire calcium signalling proteins in NSCLC. Moreover, restricting extracellular calcium in NSCLC cells restores sensitivity to EGFR-targeted therapy [270]. However, research investigating the synergistic role of EGFR and ion channels in GBM is limited. The fact that EGFR-amplified brain tumours are enriched with astrocyte-like cell states reflects the importance of maintaining ionic currents in these cells. Thus, EGFR-amplified, astrocyte-like GBMs are likely to have dysregulated ion channel expression that mirrors the highly proliferative state by increasing intracellular calcium and uncontrolled calcium release, driving proliferation, migration, and invasion. Ion channels have been investigated individually for their roles in regulating ionic cellular influxes and downstream effects on cell behaviours, such as cell growth. Although ion channels are undoubtedly involved in cellular plasticity, particularly in the neuronal setting, their contributions to maintaining tumour cell plasticity in high grade glioma has been given little research attention. More specifically, more studies are needed to elucidate the role of ANOs in regulating localised calcium signalling in synergy with calcium channels, like IP_3_R and TRP channels, and calcium-regulated growth factor receptors, such as EGFR. These interactions likely play an underestimated role in driving cancer cell progression and drug resistance, and ultimately may be responsible for the highly plastic and therapeutically challenging nature of GBM. This gap signifies an exciting avenue of research and may lead to new therapeutic discoveries that address the unmet clinical need for improved treatment options for GBM patients.

## Figures and Tables

**Figure 1 cancers-14-05932-f001:**
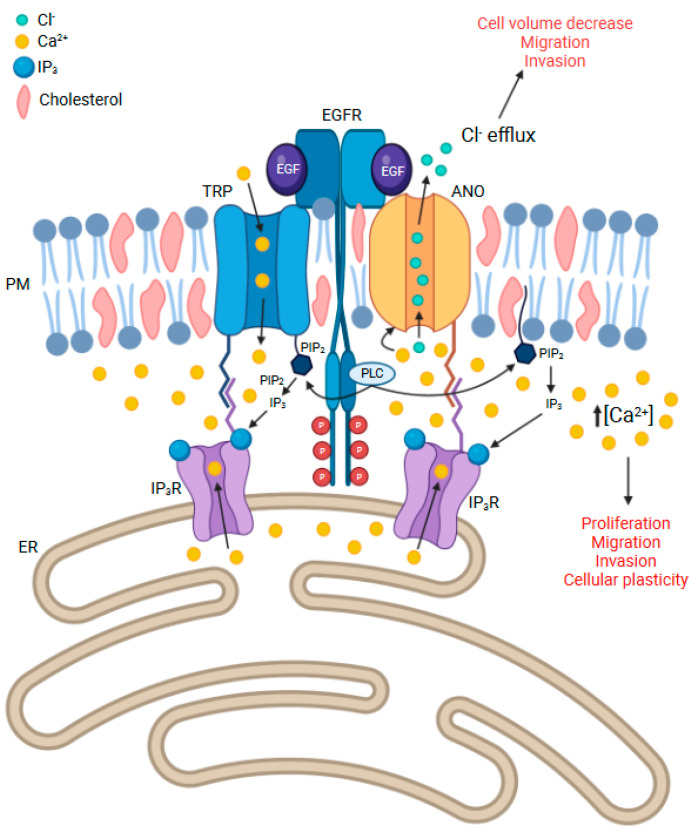
Anoctamin function in calcium-mediated cell signalling. In this model, ANO channels exist in lipid rafts within the plasma membrane, bound to receptor proteins such as EGFR, and other ion channels, such as TRP channels, for intracellular signalling. Through this interaction, calcium influx into the cell may activate ANO channels, stimulating chloride efflux and regulated volume decrease, leading to cell migration. Stimulation and activation of EGFR leads to PLC activation and cleavage of PIP_2_ to form IP_3_. IP_3_ activates the IP_3_R on the endoplasmic reticulum, causing the release of ER calcium stores. Through TRP channels and IP3R activation, localised calcium oscillations, and EGFR activation, may contribute to intracellular signalling and cellular plasticity. In an alternative model, ANO/TRP complexes may also serve to stabilise KID-EGFR as a pro-survival mechanism, resulting in calcium remodelling to sustain activation of migration and invasion pathways. ANO, anoctamin; Ca^2+^, calcium; Cl^−^, chloride; EGFR, epidermal growth factor receptor; ER, endoplasmic reticulum; IP_3_, inositol 1,4,5-trisphosphate; IP_3_R, inositol 1,4,5-trisphosphate receptor; KID, kinase independent; P, phosphate; PIP_2_, phosphatidylinositol 4,5-bisphosphate; PLC, phosphoinositide phospholipase C; PM, plasma membrane; TRP, transient receptor potential channel.

**Table 1 cancers-14-05932-t001:** EGFR-targeted therapy in GBM clinical trials.

Drug Class	Compound	Trial	Patients	Intended Therapy	Survival Outcomes	Other Comments	Reference
1st generation EGFR SMI	Gefitinib	I/II	ND	Combination with RT	OS: 11.5 months; no OS benefit vs. RT alone.	Younger age correlates with better outcome; EGFR expression no prognostic value for treatment.	[53]
		II	ND	Adjuvant (post-RT)	No difference in OS/PFS; only patients with AE demonstrated improved OS.	Clinical outcomes not affected by EGFR status.	[50]
		II	Recurrent	Monotherapy	EFS: 17 weeks; OS: 39.4 weeks; 1-year survival probability: 35.6%.	Well tolerated; clinical outcomes not affected by EGFR status.	[51]
		II	Recurrent	Monotherapy	17.9% patients showed disease stabilisation; OS: 24.6 weeks; PFS(6): 14.3%; PFS(12): 7.1%.	Limited activity; EGFR status & p-Akt expression not predictive of drug activity.	[52]
	Erlotinib	I	ND	Combination with RT	TTP: 26 weeks; OS: 55 weeks; progression in 84% of patients.	MTD not reached.	[60]
		I/II	ND	Combination with RT and TMZ	OS: 15.3 months; no benefit in OS vs. TMZ controls.	AE (grade 3/4); EGFR/PTEN/p53 status not predictive of survival.	[55]
		I/II	Recurrent	Combination with temsirolimus (mTor inhibitor)	PFS6: 13%.	MTD: 15 mg temsirolium weekly + 150 mg erlotinib daily; MTD lower than expected due to increased toxicity; EGFR status not correlated with PFS.	[57]
		I/II	Recurrent	Monotherapy	PFS: 1.9 months; OS: 6.9 months; all patients showed disease progression whilst receiving treatment	90% patients with severe AE.	[61]
		II	ND	Combination with RT and TMZ	PFS: 2.8 months; OS: 8.6 months; 4 (11.1%) treatment-related deaths.	Not efficacious with unacceptable toxicity; trial terminated after accrual of 27 patients.	[54]
		II	ND	Combination with RT, TMZ, and bevacizumab (VEGF inhibitor)	OS: 19.8 months; PFS: 13.5 months.	Well tolerated; improved PFS but not OS.	[63]
		II	Recurrent	Monotherapy	PFS(6): 11.4% (control 24%).	Well tolerated; EGFR and pAkt status not correlated with outcomes.	[56]
		II	Recurrent	Monotherapy	OR: 6.3%, response duration: 7 months, 6PFS: 20%; OS: 9.7 months.	Outcomes not related to EGFR amplification/EIAED status.	[59]
		II	Recurrent	Monotherapy	PFS(6): 3%; PFS: 2 months; OS at 12 months: 57%.	1° toxicity dermatologic; no effect on EGFR signalling; rash development in cycle 1 correlated with survival.	[62]
	Lapatinib	I/II	Recurrent	Combination with pazopanib (antiangiogenic)	MTR not reached; PFS6: 0%–PTEN/EGFRvIII-positive, 15%–PTEN/EGFRvIII-negative.	Early termination from poor survival; PK data determined exposure to lapatinib subtherapeutic.	[58]
2nd generation EGFR SMI	Afatinib	I/II	Recurrent	Monotherapy compared to combination with TMZ	PFS6: 3% MT, 10% with TMZ; 1 partial response with MT, 2 with TMZ.	MTD: 40mg/day; PFS longer in EGFRvIII-positive tumours vs. EGFRvIII-negative tumours.	[65]
	Dacomitinib	II	Recurrent	Monotherapy	PFS6: 10.6%; PFS: 2.7 months; OS: 7.4 months; 1 complete response; 2 (4.1%) responses.		[66]
		II	EGFR gene amplification	Monotherapy	PFS12: 8.9%; 14.3% of Px experienced clinical benefit.	EGFRvIII/EGFR ECD missense mutation not associated with clinical benefit.	[67]
3rd generation EGFR SMI	Osimertinib	II	EGFR gene amplification	Monotherapy	No results published.		NCT03732352
Anti-EGFR antibodies	Cetuximab	II	Recurrent	Combination bevacizumab and irinotecan (chemotherapy)	PFS6: 30%; OS: 29 weeks; efficacy not superior versus bevacizumab/irinotecan alone.	RR: 2 Px complete, 9 partial; well tolerated except for skin toxicity.	[72]
		II	Recurrent	Monotherapy	TTP: 19.9 months; PFS < 6 months; OS: 5 months.	No significant correlation between response, survival, EGFR amplification.	[73]
	Nimotuzumab	I/II	Malignant gliomas	Combination with radiochemotherapy	OS: 10.4 months (control 10.5 months); 1 year survival rates: 81.3% (69.1%); RR: 7.0% (52.2%).	Differences not significant; trend towards improved treatment efficacy.	[74]
		II	ND	Combination with RT and TMZ	PFS: 10 months; OS: 15.9 months; PFS6: 69.2%.	No correlation between efficacy & EGFR expression; survival similar to historical data of standard therapy.	[75]
		III	ND	Combination with radiochemotherapy	PFS12: 25.6% (control 20.3%); PFS; 5.6 months (4.0 months); OS: 19.5 months with residual tumour, 23.3 months without.	WT; EGFR amplification did not correlate with efficacy.	[76]
	Depatux-m	I	ND; recurrent	Monotherapy compared to combination with TMZ	PFS6: 30.8%; OS: 10.7 months.	MTD: 1.5 mg/kg with TMZ, not reached as MT; RP2D: 1.25 mg/kg.	[77]
		II	Recurrent; EGFR amplification	Monotherapy compared to combination with TMZ	PFS: 1.9 months (Depatux-m) vs. 2.7 months (Depatux-m + TMZ); OS: 7.9 months vs. 9.6 months.		[78]
		Observational	Recurrent	Combination with TMZ	OS: 9.04 months; OS(12): 37%.	MGMT methylation status only factor significantly associated with survival; moderate, manageable toxicity.	[79]
	ABT-414	II	ND	Combination with RT and TMZ	No results published.		NCT02573324
EGFRvIII peptide vaccine	Rindopepimut	II	ND	Combination with TMZ	PFS(6): 66%; OS: 21.8 months; OS(36): 26%.	Well tolerated; EGFRvIII eliminated in 4/6 (67%) of tumour samples after >3 months therapy.	[80]
		III	ND; EGFRvIII mutant	Combination with TMZ	No significant difference in OS; OS: 20.1 months vs. 20.0 months (control).		[81]
CAR T cell therapy	NA	I	Recurrent; EGFRvII mutant	Monotherapy	PFS: 1.3 months; OS: 6.9 months.	Persistence of CAR+ cells correlated with dose but no objective responses.	[82]
		I	Recurrent	Monotherapy	No results published.		NCT03618381

AE, adverse effect; BCNU, carmustine; CAR, chimeric antigen receptor; CO, clinical outcomes; CT, chemotherapy; DE, dose escalation; DLT, dose limiting toxicity; ECD, extracellular domain; EGFR, epidermal growth factor receptor; EIAC, enzyme-inducing anticonvulsant; MGMT, O6-methylguanine-DNA-methyltransferase; MT, monotherapy; MTD, maximum tolerated dose; mTOR, mammalian target of rapamycin; MTR, maximum treatment regimen; NA, not available; ND, newly diagnosed; OR, overall response; OS, average overall survival (months); OS(12/36), overall survival at 12 or 36 months (% of patients); p-Akt, phospho-protein kinase B; PEP, primary end point; PFS, average progression free survival; PFS(6/12), progression free survival at 6 or 12 months; PK, pharmacokinetic; PTEN, phosphate and tensin homolog; Px, patient; RP2D, recommended phase 2 dose; RR, response rate; RT, radiotherapy; SMI, small molecule inhibitor; TMZ, temozolomide; TTP, time to progression; VEGF, vascular endothelial growth factor; WT, well tolerated.

**Table 2 cancers-14-05932-t002:** ANOs in cancer.

ANO	Role	Cancer Type	Comments/Mechanism	Reference
ANO1	Proliferation	OSCC	OE; PS; promotes MVA pathway.	[134]
		Breast	OE; upregulates EGFR/HER2 expression; activates EGFR and CAMK signalling.	[135,136]
		Glioma	OE; upregulates NF-κB pathway.	[137]
		Colorectal	OE.	[138]
		HNSCC	OE; PS; binds and stabilises EGFR.	[139]
	Metastasis	Gastrointestinal	OE; PS.	[140]
		HNSCC	UE; switch between proliferation and metastasis; dependent on promoter methylation status.	[141]
	Invasion, migration	Glioblastoma	OE; binds and stabilises EGFRvIII; activates MAPK/PI3K signalling pathways.	[142]
ANO5	Proliferation	Pancreatic	OE; PS.	[143]
	Migration	Thyroid	UE; activates JAK/STAT3 pathway.	[144]
	Proliferation, migration	Osteosarcoma	OE; promotes NELL1/2 degradation.	[145]
ANO6	Proliferation, invasion	Glioma	OE; PS; via ERK activation.	[146]
ANO7	Differentiation	Prostate	UE; PS; interacts with PTEN/Akt pathway.	[147]
ANO9	Proliferation	Pancreatic	OE; PS; activates ERK/EGFR signalling.	[148]
		Colorectal carcinoma	UE; PS; inhibits ANO1 activity.	[149]

Akt, protein kinase B; CAMK, calmodulin-dependent protein kinase; EGFR, epidermal growth factor receptor; HER2, human epidermal growth factor receptor 2; HNSCC, squamous cell carcinoma of the head and neck; JAK, Janus kinase; MAPK, mitogen-activated protein kinase; MVA, mevalonate acid; NELL, Nel-like proteins; NF-κB, nuclear factor kappa B; OE, overexpressed; OSCC, oral squamous cell carcinoma; PI3K, phosphoinositide 3-kinase; PS, correlated with poorer survival; PTEN, phosphatase and tensin homolog; STAT, signal transducer and activator of transcription; UE, under-expressed.

**Table 3 cancers-14-05932-t003:** ANO inhibitors in cancer.

Compound	ANO Target	Condition	Effect of Inhibition	Reference
CaCC_inh_-AN01	ANO1 ANO2	Prostate cancer	Inhibited proliferation and induced apoptosis	[172]
		Colon cancer		
		GBM	Suppressed GSC activities, reduced expression of stemness-related factors, and reduced EGFRvII signalling	[142]
		HNSCC	Reduced cell viability via reduced EGFR activity and sensitised cells to EGFR-targeted therapy	[139,170]
DES	ANO1 ANO2	NSCLC	Inhibited cell viability and migration through reduced activation of EGFR	[171]
3n	ANO1 ANO2	Prostate cancer	Reduced cancer cell viability and induced apoptosis via caspase 3 activation and PARP cleavage	[173]
2-aminothiophene-3-carboxamide	ANO1	GBM	Suppressed proliferation, migration, and invasion of GBM cells	[174]
Cepharanthine	ANO1 ANO2 ANO6	Lung adenocarcinoma	Inhibited cell proliferation and migration, induced apoptosis, and reduced tumour growth	[175]
NFA	ANO1 ANO2 ANO6	Hepatoma	Blocked cell cycle progression and decreased intracellular Ca^2+^	[176]
		NPC	Inhibited proliferation, migration, and invasion through reduced ERK	[177]
NPPB	ANO1 ANO2 ANO6	Glioma	NPPB conjugated to TMZ inhibited cell proliferation, migration, and invasion	[178]
		OSCC	Induced EMT via Wnt/ß-catenin signalling	[179]
Niclosamide	ANO1 ANO6	GBM	Reduced GBM malignancy via reduced Wnt, Notch, mTOR, and NF-κB signalling	[180,181,182]
Gallotannin	ANO1 ANO2	GBM	Inhibited cell proliferation	[183]
		Colorectal cancer	Inhibited lung metastasis; regulated PI3K/Akt/mTOR, AMPK signalling pathways; downregulated mesenchymal marker expression; induced senescence	[184,185,186]
		Breast cancer	Inhibited proliferation via increased Chk2 phosphorylation	[187]
		Prostate cancer	Induced apoptosis via reduced Mcl-1 signalling and activation of procaspase 9/3 expression	[188]
		Liver cancer	Reduce cell viability via increased SIRT1, mTOR, and activated AMPK	[189]

Akt, protein kinase B; AMPK, AMP-activated protein kinase; Chk2, checkpoint kinase 2; DES, Diethylstilbestrol; EGFR, epidermal growth factor receptor; EMT, epithelial to mesenchymal transition; GBM, glioblastoma; GSC, glioma stem cells; HNSCC, head and neck squamous cell carcinoma; Mcl-1, myeloid cell leukemia 1; mTOR, mammalian target of rapamycin; NF-κB, nuclear factor kappa B; NPC, nasopharyngeal carcinoma; NPPB, 5-nitro-2-(3-phenylpropylamino) benzoic acid; NSCLC, non-small cell lung cancer; OSCC, oral squamous cell carcinoma; PARP, polyadenoside diphosphate-ribose polymerase; PI3K, phosphoinositide 3-kinase; SIRT1, sirtuin 1; TMZ, temozolomide; Wnt, wingless-related integration site; 3n, (E)-1-(7,7-dimethyl-7H-furo [2,3-f]chromen-2-yl)-3-(1H-pyrrol-2-yl)prop-2-en-1-one.

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
