# Peer review of "Anoctamins and Calcium Signalling: An Obstacle to EGFR Targeted Therapy in Glioblastoma?"

_cancers, 2022, doi:10.3390/cancers14235932_

Round 1

Reviewer 1 Report

Overview and general recommendation:

In the manuscript, the authors introduce that epidermal growth factor receptors (EGFR) are drug targets in glioblastoma treatment. Many inhibitors and antibodies of EGFR show anit-tumor effect, but can’t help in improving glioblastoma patients’ outcomes. Research in ion channels may help in regulating cellular plasticity in glioblastoma, thus improving the effect of inhibitors and antibodies of EGFR in the clinical trials.

First they show the EGFR signaling in Glioblastoma multiforme (GBM) and in most GBM, expression of EGFR is upregulated. Then they summarize some clinical results of inhibitors, antibodies and other EGFR-targeted agents. None of them shows promising result in clinical trials. The authors summarize the role of EGFR in tumorigenesis in GBM and find that ion channels may contribute to GBM. In this review, the authors concentrate in a calcium-activated chloride channel called anoctamins (ANOs) and introduce the role of ANOs in cell migration and tumorigenesis, especially in glioma. Then they introduce the correlation of calcium channels ANOs and EGFR-related signaling.

I find the paper is well organized. The author performed detailed background research and all the parts are organized in a logic way. The figures are quite clear to show the model proposed by the authors. I have no doubt about recommending it for publication.

Minor comments:

1.      Page16, the font of first part seems to be different.

Author Response

Thank you for your kind words regarding our review.

Reviewer Point 1: Page16, the font of first part seems to be different.

Response: We apologise for the formatting error and thank you for bringing it to our attention. We have now amended the font on page 16 for consistency with the MDPI template.

Reviewer 2 Report

In the current manuscript, entitled " Anoctamins and calcium signalling: an obstacle to EGFR targeted therapy in glioblastoma?”, Dewdney et al., studied why many inhibitors and antibodies for EGFR have demonstrated promising anti-tumor effects in preclinical models, but they have failed to improve outcomes for glioblastoma patients in clinical trials. They investigated the potential involvement of a class of calcium-activated chloride channels called anoctamins in regulating calcium-mediated signalling pathways, such as EGF signalling in brain cancer.

In general, this study provided preeminent and updated source of information about EGFR signaling underlying GBM progression, and summarized the relationship between calcium-activated chloride channels and EGF signaling. They point out an exciting research avenue in GBM that is the relationship between calcium channels, like ANO, IP3R and TRP channels, and calcium-regulated growth factor receptors, such as EGFR.

Minor points:

1. Does the author add some information about heterogeneity of EGFR in GBM? Like what cause this heterogeneity?

2. On page 16, the font is changed.

Author Response

We thank the reviewer for their comments. Please see below for our point-by-point response.

Reviewer Point 1:  Does the author add some information about heterogeneity of EGFR in GBM? Like what cause this heterogeneity?

Response: Like most cancers, intratumoural heterogeneity is a common feature in glioblastoma that develops as the tumour progresses. Although not fully understood, there are a few key factors that may contribute to EGFRvIII heterogeneity. First, EGFRvIII drives the growth of EGFR amplified cells, thus there is not a strong selection for EGFRvIII cells in the whole tumour. Second, EGFRvIII expression is a late event, so many other clones are already established before its appearance. Finally, unequal segregation of the extrachromosomal DNA fragments during cell division may further contribute to EGFRvIII heterogeneity. We have now added this information to clarify (please see revised manuscript section 2, page 3). 

Reviewer Point 2: On page 16, the font is changed.

Response: Thank you for noting our formatting error. We have now amended the font so that it is consistent with the rest of the MDPI template.